# Anti-Inflammatory and Wound Healing Properties of Leaf and Rhizome Extracts from the Medicinal Plant *Peucedanum* *ostruthium* (L.) W. D. J. Koch

**DOI:** 10.3390/molecules27134271

**Published:** 2022-07-02

**Authors:** Cristina Danna, Miriam Bazzicalupo, Mariarosaria Ingegneri, Antonella Smeriglio, Domenico Trombetta, Bruno Burlando, Laura Cornara

**Affiliations:** 1Department of Earth, Environment and Life Sciences (DISTAV), University of Genova, Corso Europa 26, 16132 Genova, Italy; cristina.danna@edu.unige.it (C.D.); cornaral@gmail.com (L.C.); 2CREA—Research Centre for Vegetable and Ornamental Crops, Council for Agricultural Research and Economics, Corso Inglesi 508, 18038 Sanremo, Italy; miriam.bazzicalupo@gmail.com; 3Department of Chemical, Biological, Pharmaceutical and Enviromental Sciences (ChiBioFarAm), University of Messina, Viale Ferdinando Stagno d’Alcontres 31, 98166 Messina, Italy; mariarosaria.ingegneri@unime.it (M.I.); antonella.smeriglio@unime.it (A.S.); domenico.trombetta@unime.it (D.T.); 4Department of Pharmacy—DIFAR, University of Genova, Viale Benedetto XV 3, 16132 Genova, Italy

**Keywords:** plants traditional use, leaf extract, rhizome extract, micromorphology, phytochemical characterization, antioxidant activity, anti-inflammatory activity, wound-healing activity

## Abstract

*Peucedanum* *ostruthium* (L.) W. D. J. Koch (Apiaceae) is a worldwide perennial herb native to the mountains of central Southern Europe. The rhizome has a long tradition in popular medicine, while ethnobotanical surveys have revealed local uses of leaves for superficial injuries. To experimentally validate these uses, plant material was collected in the Gran Paradiso National Park, Aosta Valley, Italy, and the rhizome and leaves were micromorphologically and phytochemically characterized. Polyphenol-enriched hydroalcoholic rhizome and leaf extracts, used in cell-free assays, showed strong and concentration-dependent antioxidant and anti-inflammatory activities. In vitro tests revealed cyclooxygenase and lipoxygenase inhibition by the leaf extract, while the rhizome extract induced only lipoxygenase inhibition. MTT assays on HaCaT keratinocytes and L929 fibroblasts showed low cytotoxicity of extracts. In vitro scratch wound test on HaCaT resulted in a strong induction of wound closure with the leaf extract, while the effect of the rhizome extract was lower. The same test on L929 cells showed similar wound closure induction with both extracts. The results confirmed the traditional medicinal uses of the rhizome as an anti-inflammatory and wound healing remedy for superficial injuries but also highlighted that the leaves can be exploited for these purposes with equal or superior effectiveness.

## 1. Introduction

*Peucedanum* *ostruthium* (L.) W. D. J. Koch (syn. *Imperatoria ostruthium* L.), commonly known as masterwort, is a rhizomatous perennial species belonging to the Apiaceae family. Native to the mountains of central Southern Europe, it is widespread around the world and generally grows in rivers banks and wet grassy and anthropic areas (Figure 1). The rhizome has a long tradition for liqueur production and as a popular medicine, to such an extent that during the 19th century the plant was known as ‘*Divinum remedium*’ (divine remedy) [1]. Records from historical and modern ethnobotanical studies have reported a long list of popular uses of the plant [2,3,4]. More specifically, it has been employed as a stimulant, stomachic, and diuretic for rheumatic, chronic inflammatory, and musculoskeletal diseases, as well as for skin problems, typhoid fever, paralytic conditions, and delirium tremens [5,6].

Experimental studies have been conducted to explore the alleged therapeutic properties of the plant, revealing the presence of different bioactive coumarin compounds. The major components of the essential oil (EO) from the rhizome are sabinene and 4-terpineol, while β-caryophyllene and α-humulene are dominant in the leaf EO [7]. Anti-inflammatory activity has been found in an ethanolic extract used on a carrageenan-induced rat paw edema model [8]. The main compound isolated from this extract, 6-(3-carboxybut-2-enyl)-7-hydroxycoumarin, has shown significant inhibition of edema, while the extract and the isolated coumarin have decreased prostaglandin release in stimulated rabbit ears. In addition, both extract and coumarin have shown antipyretic effects on rats, with coumarin being more potent than acetylsalicylic acid and indomethacin [9]. Another study has shown an anti-inflammatory mechanism of a rhizome extract based on the inhibition of the NF-KB pathway [10]. The coumarin compound, ostruthol, has been found to strongly inhibit acetylcholinesterase activity in bioautographic TLC assays [11]. The coumarin ostruthin isolated from the rhizome dichloromethane extract has been identified as an inhibitor of vascular smooth muscle cell proliferation [12]. Ostruthin has also exhibited inhibitory activities against the rapidly growing *Mycobacterium fortuitum*, *M. aurum*, *M. phlei*, and *M. smegmatis* [13]. Similarly, the coumarin compound oxypeucedanin hydrate from the rhizome ethyl acetate extract has exerted strong antibacterial activity on *Bacillus cereus* but not on *Escherichia coli* or *Staphylococcus aureus* [5].

The rhizome is the part of the plant officially recognized in traditional medicine, as shown by its inclusion in the European Belfrit list of botanicals [14]. Among its different uses, the rhizome is known as a remedy for superficial injuries and wounds, but a similar use of the leaves has been reported among the local population [2]. This suggests new possibilities of exploitation for medicinal purposes and a reevaluation of the leaves of this plant, which are generally discharged during rhizome harvesting for medicinal purposes.

In this study, we provide experimental confirmations of the therapeutic properties of the *P.*
*ostruthium* rhizome extract, as suggested by traditional usage. Moreover, we confirm the indications of empirical evidence suggesting that the leaves can also be used as a medicinal remedy, thereby opening the way to more intensive exploitation of the plant. By using leaf and rhizome hydroalcoholic extracts enriched in polyphenols, we have studied different bioactivities through in vitro cell-free and cell-based assays in terms of antioxidant, anti-inflammatory, and wound healing properties.

## 2. Results

### 2.1. Micromorphological Analysis

Light microscopy observations of leaf transversal sections show the proximity of the midvein vascular bundles surrounded by secretory channels that produce essential oils. The channels are located above and below or on one side of vascular bundles and protected by collenchyma protrusions (Figure 2A). Polyphenol/tannin cells in the parenchyma are highlighted by TBO staining appearing in blue/green (Figure 2B). In scanning electron microscopy (SEM) micrographs of non-glandular trichomes emerging on the leaf surface in correspondence with collenchyma protrusions are well visible (Figure 2C). In addition, in a transversal section of the dorsoventral leaf anatomy, the palisade and spongy parenchyma can be seen (Figure 2D).

Light microscope analysis of the rhizome reveals a structure consisting of different vascular bundles (Figure 3A), where phloem and xylem are about the same size. The stratified phloem shows little or no lignified fiber, while the xylem is interspersed with sclerenchyma fibers with extremely thick and lignified walls (Figure 3B). SEM micrographs highlight the circular shape of the rhizome with a secondary structure of the fascicular type, in which a ring of 30–70 collateral vascular bundles surrounds the central medulla and are separated by multiseriated medullary rays (Figure 3C). Large secretory ducts are located both in the cortex and in the peripheral zone of the medulla. The rhizome is rich in amiliferous parenchyma and around the secretory channels the parenchymatous cells are oriented in concentric layers (Figure 3D). Externally, the rhizome is covered by cork layers and a multi-layered phelloderm.

### 2.2. Phytochemical Analyses

The phytochemical screening has shown statistically significant differences between the leaf hydroalcoholic extract (LE) and the rhizome hydroalcoholic extract (RE) investigated in terms of flavonoids, flavanols, and proanthocyanidin content; the last two parameters are also useful to calculate the polymerization index. On the contrary, no statistically significant difference was found in terms of total phenols (Table 1).

LE showed the highest content of flavonoids (about four times the content of RE) and proanthocyanidins (about 26 times), showing a very high polymerization index, indicating a preponderance of hydrolysable tannins. These data have been confirmed by LC-DAD-ESI-MS analysis (Table 2).

As shown in Table 2, among the major components identified, LE showed the greatest content of flavonoids (76.23% vs. 1.31%), followed by phenolic acids and coumarins. Conversely, RE showed a higher content of phenolic acids and coumarins than LE, 53.32% vs. 16.97%, and 45.37% vs. 6.80%, respectively, and a very low content of flavonoids (1.31%). Kaempferol 3-*O*-acetyl-glucoside (501.24 ± 0.66 mg/100 g DE) was the most abundant component identified in LE, followed by quercetin-3-*O*-(6″acetyl-glucoside) (138.52 ± 1.88 mg/100 g DE), 4-*O*-caffeoylquinic acid (135.01 ± 0.57 mg/100 g DE), quercetin-3-*O*-rutinoside (95.28 ± 0.84 mg/100 g DE) and oxypeucedanin-hexoside (46.98 ± 0.42 mg/100 g DE). RE showed instead a completely different phytochemical profile with a preponderance of phenolic acids and coumarins. Ostruthin (281.88 ± 2.24 mg/100 g DE) was the most abundant compound, followed by 3-*O*-caffeoylquinic acid (195.55 ± 1.67 mg/100 g DE), 5-*O*-caffeoylquinic acid (114.36 ± 1.44), 5-*O*-Feruloylquinic acid (59.80 ± 0.05) and isoimperatorin (29.55 ± 0.08).

### 2.3. Antioxidant and Anti-Inflammatory Activities

The antioxidant and anti-inflammatory activities of LE and RE were firstly screened in vitro using cell-free tests based on different environments and reaction mechanisms (Table 3). Results were expressed as half-maximal inhibitory concentration (IC_50_, µg/mL) with confidence limits (C.L.). LE showed the best biological activity in all assays accordingly to its higher content in flavonoids and phenolic acids, which are known to be involved in free-radical scavenging processes and chelating activity as enzyme-cofactors. The same trend was observed also in the two anti-inflammatory tests, which evaluate the ability of the extracts to counteract heat-induced bovine serum albumin (BSA) denaturation and to inhibit protease activity, playing a pivotal role in many inflammatory diseases.

The in vitro inhibitory effects of LE and RE on pro-inflammatory enzymes, such as cyclooxygenase (COX-2) and lipoxygenase (LOX), showed about 65% inhibition of COX-2 by LE at 150 µg/mL, followed by a slight reduction in the effect at 300 µg/mL.

Conversely, no inhibitory effect was observed with RE (Table 4). The concentration-dependent inhibition of LOX was observed for both LE and RE, but with a significantly stronger effect of LE at 150 µg/mL. Overall, LE was significantly effective on both enzymes at a sub-cytotoxic concentration of 150 µg/mL, while RE was only effective on lipoxygenase at 300 µg/mL.

### 2.4. Cell Viability and Wound-Healing Activity

The effects of RE and LE on cell viability have been evaluated by the MTT assay using HaCaT keratinocytes and L929 fibroblasts as skin cell experimental models. At the endpoint of 24 h, negligible cytotoxic effects were found with both extracts, with the exception of 1000 µg/mL RE. At 48 h, a dose-dependent decrease of cell viability for increasing extract concentrations was observed, with a steeper slope for exposure to RE in both cell types, revealing a more hypersensitive response (Figure 4). Values of median inhibitory concentrations (IC_50_) and threshold effective concentrations (IC_05_) at 48 h indicate low cytotoxicity of the extracts, with the lowest effects recorded for LE on L929 cells and a minimum threshold value of about 250 µg/mL for the effect of the same extract on HaCaT (Table 5).

The effects of extracts on skin cell migration were evaluated by in vitro scratch wound assays on HaCaT and L929 monolayers. The ability of the epithelially arranged HaCaT keratinocytes to close the wound while maintaining clearly identifiable wound edges allowed the evaluation of the healing process by measuring the wound width just after the wounding and 24 h post wounding. Conversely, wound closure by L929 fibroblasts was much more irregular and had to be evaluated by measuring the cell density within the wound space at the same time points. The LE induced on HaCaT a rapid wound closure at the lowest concentration of 15 µg/mL, superior to that induced by the positive control allantoin, followed by a significant decrease of the effect at higher concentrations (Figure 5). A similar trend was observed on L929 cells, but in this case, the maximum effect of the 15 µg/mL concentration was significantly lower with respect to allantoin (Figure 5). The effect on HaCaT of RE was lower than that of LE and started from a concentration of 70 µg/mL, whereas on L929, the effect of RE was overall similar to that of LE (Figure 5).

## 3. Discussion

Traditional medicine reports the use of *P. ostruthium* for the preparation of medicaments such as tisanes, decoctions, or liqueurs. Considering this, to optimize the extraction of bioactive compounds from leaves and rhizomes, a hydroalcoholic mixture was used, obtaining a high extraction yield comparable to others previously reported (Palmioli et al., 2019) [15].

According to previous studies [15,16,17], LC-ESI-DAD-MS analysis allowed identifying 23 main components in *P. ostruthium* LE and RE, belonging to different polyphenol classes, such as coumarins, phenolic acids, and flavonoids. However, important differences in terms of secondary metabolites were found. These differences could be ascribed to the climatic and environmental features of the sampling sites [18] but also to the extraction method adopted [15,16,17]. Regarding LE, which has been poorly investigated to date, these differences stand out mainly in terms of the relative abundance of flavonoids, especially kaempferol and quercetin derivatives, with respect to phenolic acids such as caffeoyl and feruloylquinic derivatives [17]. Considering RE, according to our data, two previous studies identified the caffeoyl derivatives as the preponderant class, followed by coumarins [15,16]. Moreover, even within the same class of compounds, significant differences were recorded in terms of relative abundance. In the present study, a preponderance of ostruthin was highlighted, while previous studies reported a preponderance of oxypeucedanin and derivatives [17].

Considering COX and LOX enzymes, a previous study reported their inhibition by *P. ostruthium* rhizome extracts [8]. However, our data indicate a stronger inhibitory activity by LE with respect to RE. Moreover, a stronger effect of LE than RE was also found in the wound healing assay on HaCaT cells, and this finding agrees with a study on a skin ulcer model in rats treated with an ointment containing also a *Peucedanum* leaf extract [19].

What emerges for the first time from this study is that leaves and rhizomes are sharply different from the phytochemical point of view. The RE contains about seven times and three times more coumarins and phenolic acids than LE, and a very low content of flavonoids (1.31% in RE vs. 76.23% in LE), represented exclusively by hesperidin, as previously identified [15,16]. These substantial differences translate into different biological activities, with LE proving to be the most promising for antioxidant, anti-inflammatory, and wound healing activities. This is in contrast with previous ideas that the health effects of these extracts are mainly attributable to phenolic acids, i.e., caffeoyl and feruloylquinic derivatives. Flavonoids are probably the discriminating factor in terms of biological activity, although a synergistic effect of the two classes of compounds cannot be excluded.

In RE, the two major constituents are the coumarins ostruthin and isoimperatorin, which are known for cytotoxic and antiproliferative activities [20,21] and could play a role in the slightly stronger cytotoxic effect of RE with respect to LE on both HaCaT and L929 cells. In addition, a role in the RE inhibition of proinflammatory enzymes could be ascribed to caffeoyl and feruloylquinic derivatives [22]. In LE, the much higher flavonoid content with respect to RE could explain the stronger inhibition of proinflammatory enzymes and wound healing activity. Quercetin, abundant in LE in glycosylated form, has been shown to promote wound healing in different studies, including in vitro on HaCaT cells through epithelial–mesenchymal transition [23]. Quercetin has also been reported to inhibit both COX and LOX activities [24] and could be responsible for the stronger inhibition of these enzymes observed in LE with respect to RE.

Finally, it should be emphasized that the phytochemical profile of LE is also favorable from the toxicological point of view, as it contains much fewer coumarins than RE, paving the way to a very promising use of this extract in the nutraceutical and cosmeceutical fields. The scalability to industrial production for the plant has already been achieved, and the rhizome extract is on the market (e.g., see https://www.minardierbe.it, accessed on 28 June 2022). Therefore, the introduction of the leaf extract as an herbal healthcare product would result in a considerably improved exploitation of this plant with positive economic and environmental outcomes.

## 4. Materials and Methods

### 4.1. Plant Material

Plant material was collected at the Gran Paradiso National Park, Buthier, Cogne, Aosta Valley, Italy (alt: 1555 m a.s.l.; lat. 45.603671; long. 7.3494468). Permissions for plant sampling were obtained from the National Park Authority (Ente Parco Nazionale Gran Paradiso, Torino, Italy) n. 1884, 6 September 2020; n. 2432/2020, 7 October 2020; n. 2959/2020 8 October 2020. Voucher specimens were deposited at the Ethnobotanical Herbarium of the Gran Paradiso National Park, Paradisia Alpine Botanic Garden (Valnontey, Cogne-AO, Italy) (P.ost. HBPNGP_ETN). The nomenclature follows the Plants of the World Online, Kew Science classification, available at http://www.plantsoftheworldonline.org/ (accessed on 28 June 2022), and the corresponding synonymous was added, according to the IPFI: Index Plantarum, available online at https://www.actaplantarum.org/flora/flora.php (accessed on 28 June 2022).

### 4.2. Chemicals and Cells

Reagents were purchased from Sigma-Aldrich (Milan, Italy) unless otherwise specified. Reference standards (purity ≥ 98%) of compounds reported in Table 2 were purchased from Extrasynthase (Genay, France) and Merck (Darmstadt, Germany).

The HaCaT human keratinocyte cell line was obtained from DKFZ, Deutsches Krebsforschungszentrum, Heidelberg, Germany [25], while L929 cells were from the Tissue Bank of the IRCCS San Martino Hospital (Genova, Italy). The cell lines were used at passage levels around 50 (HaCaT) and 130 (L929).

### 4.3. Micromorphological Analyses

Fresh leaves and rhizomes were hand-cut with a razor blade, and cross sections were observed using a transmission light Leica DM 2000 microscope equipped with a DFC 320 camera (Leica Microsystems, Wetzlar, Germany). Lignin was stained with a phloroglucinol-HCl test, while phenolic substances were stained with toluidine blue O (TBO) [26].

For SEM analyses, leaves and rhizome were sectioned (about 2 cm^2^) and incubated overnight at 4 °C in 70% ethanol/FineFIX solution (Milestone s.r.l., Bergamo, Italy) [27]. Samples were then dehydrated with an ethanol series, critical point dried (K850CPD 2 M Strumenti s.r.l., Roma, Italy), placed on aluminum stubs, and sputter-coated with 20 nm gold. Observations were carried out under a Vega3-Tescan LMU SEM microscope (TescanUSA Inc., Cranberry Twp, PA, USA) at an accelerating voltage of 20 kV.

### 4.4. Sample Preparation

Fresh leaves and rhizomes were powdered by a blade mill (IKA^®^ A11, IKA^®^-Werke GmbH & Co. KG, Staufen, Germany) with liquid nitrogen to block the enzymatic activities and to preserve the native chemical features. A sample preparation procedure was developed and optimized with the aim to maximize the extraction of all classes of bioactive compounds (coumarins, phenolic acids, and flavonoids), following the traditional use of the plant and using food-grade solvents. To this end, several ethanol/water (EtOH/H_2_O) blends were tested (90:10, *v*/*v*; 80:20, *v*/*v*; 70:30, *v*/*v*; 60:40, *v*/*v;* and 50:50, *v*/*v*) at different extraction times (3, 6, and 9 h) and repetitions (1–4 times). Maximum yield in terms of bioactive compounds was obtained with 80:20, *v*/*v*, for 6 h, repeated three times (data not shown), which was chosen to proceed with the study.

Ten grams of rhizome and leaf frozen powders were extracted as reported above under continuous stirring in the dark at room temperature (RT) for 6 h. Extracts were then centrifuged at 3000× *g* for 15 min at 4 °C. Thereafter, the supernatants were collected and evaporated until dry by a rotary evaporator (Büchi R-205, Cornaredo, Italy) at 37 °C. Yields of 31.70% and 32.50% were obtained for the leaf (LE) and rhizome extract (RE), respectively. Extracts were then suspended and properly diluted in a hydroalcoholic mixture for phytochemical characterization and subsequent analyses.

### 4.5. Phytochemical Screening

#### 4.5.1. Total Phenols

Total phenols were quantified according to Smeriglio et al. [28]. Briefly, 50 µL of LE and RE (12.5–100 µg/mL) and gallic acid as the reference standard (75.0–600 µg/mL) were added to 450 µL of deionized water and 500 µL of Folin–Ciocalteu reagent. After 3 min, 500 µL of 10% sodium carbonate was added, and samples were left in the dark at RT for 1 h, vortex-mixing every 10 min. Absorbance was read at 785 nm with a UV–VIS spectrophotometer (Model UV-1601, Shimadzu, Kyoto, Japan) against a blank consisting of the same extraction hydroalcoholic mixture. Results were expressed as mg gallic acid equivalents (GAE)/100 g dry extract (DE). 

#### 4.5.2. Flavonoids

Total flavonoids were quantified according to Smeriglio et al. [29]. Briefly, 0.2 mL of LE and RE (0.125–1.0 mg/mL), as well as rutin (0.125–1.0 mg/mL) as reference compound, were mixed with 0.2 mL AlCl_3_ and 1.2 mL sodium acetate (2 mg/mL and 50 mg/mL, respectively) and incubated for 2.5 h at RT. The absorbance was recorded at 440 nm by using the same instrument and blank reported in Section 4.5.1. Results were expressed as mg rutin equivalents/100 g DE.

#### 4.5.3. Vanillin Index

Vanillin index determination was carried out according to Smeriglio et al. [30]. In brief, 2.0 mL of LE (1 mg/mL) and RE (4 mg/mL) diluted in 0.5 M H_2_SO_4_ were loaded onto a conditioned Sep-Pak C18 cartridge (Waters, Milan, Italy), washed with 2.0 mL of H_2_SO_4_ (5.0 mM) and eluted with 5.0 mL of methanol. Thereafter, 1 mL of each eluate was added to 6.0 mL of 4% vanillin methanol solution and incubated in a water bath at 20 °C for 10 min. Chloridric acid (3.0 mL) was added, and after 15 min, the absorbance was recorded at 500 nm by using the same instrument and blank reported in Section 4.5.1. Catechin was used as a reference compound (0.125–0.50 mg/mL). Results were expressed as mg catechin equivalents (CE)/100 g DE.

#### 4.5.4. Proanthocyanidins

Proanthocyanidins were quantified according to Barreca et al. [31]. In brief, 2.0 mL of LE (1 mg/mL) and RE (2 mg/mL), diluted with 0.05 M H_2_SO_4_, were loaded onto a conditioned Sep-Pak C18 cartridge (Waters, Milan, Italy). The proanthocyanidins-reach fraction obtained was eluted with methanol (3.0 mL) and collected in a 100 mL round bottom flask shielded from light containing 9.5 mL of absolute ethanol. Thereafter, 12.5 mL of FeSO_4_·7H_2_O chloridric acid solution (300 mg/L) was added and placed to reflux for 50 min. A sample for each extract, each prepared identically but without heating, was used to calculate the basal anthocyanin content of LE and RE. After cooling, the absorbance was recorded at 550 nm by using the same instrument and blank reported in Section 4.5.1. Proanthocyanidin content was expressed as 5 times the amount of cyanidin formed by means of a cyanidin chloride (ɛ = 34,700) calibration curve. Results were expressed as mg of cyanidin equivalents (CyE)/100 g DE.

### 4.6. Polyphenol Profile by RP-LC-DAD-ESI-MS Analysis

Polyphenol characterization of LE and RE was carried out by RP-LC-DAD-ESI-MS analysis according to Smeriglio et al. [32]. Chromatographic elution was carried out by a Luna Omega PS C18 column (150 mm × 2.1 mm, 5 µm; Phenomenex, Torrance, CA, USA) at 25 °C by using mobile phase 0.1% formic acid (Solvent A) and methanol (Solvent B) according to the following program: 0–3 min, 0% B; 3–9 min, 3% B; 9–24 min, 12% B; 24–30 min, 20% B; 30–33 min, 20% B; 33–43 min, 30% B; 43–63 min, 50% B; 63–66 min, 50% B; 66–76 min, 60% B; 76–81 min, 60% B; 81–86 min, 0% B and equilibrated 4 min. The injection volume was 5 µL. The UV–Vis spectra were recorded ranging from 190 to 600 nm, and chromatograms were acquired at different wavelengths (220, 260, 292, 330, and 370 nm) to identify all polyphenol classes. The experimental parameters of the mass spectrometer (ion trap, model 6320, Agilent Technologies, Santa Clara, CA, USA) operating in the negative (ESI−) and positive (ESI+) ionization mode were set as follows: 3.5 kV capillary voltage, 40 psi nebulizer (N_2_) pressure, 350 °C drying gas temperature, 9 L/min drying gas flow and 40 V skimmer voltage. Acquisition was carried out in full-scan mode (90–1000 *m*/*z*). Data were acquired by Agilent ChemStation software version B.01.03 and Agilent trap control software version 6.2.

### 4.7. Antioxidant Activity

The antioxidant and free-radical scavenging activity of *P. ostruthium* LE and RE was evaluated by several in vitro colorimetric assays based on different mechanisms and reaction environments. The results, which represent the average of three independent experiments in triplicate (*n* = 3), were expressed as the inhibition percentage (%) of the oxidative/radical activity, calculating the IC_50_ with the respective C.L. at 95% by Litchfield and Wilcoxon’s test using PHARM/PCS software version 4 (MCS Consulting, Wynnewood, PA, USA). All concentration ranges reported refer to the final concentrations of LE and RE and reference compounds in the reaction mixture.

#### 4.7.1. DPPH Assay

The DPPH radical scavenging activity was evaluated according to Smeriglio et al. [33]. In brief, 37.5 µL of LE (5.0–40 µg/mL) and RE (25–200 µg/mL) was added to fresh 1mM DPPH methanol solution, vortex-mixed for 10 s, and incubated in the dark at RT for 20 min. Absorbance was recorded at 517 nm using the same instrument and blank reported in Section 4.5.1. Trolox was used as a reference compound (0.63–5.0 µg/mL).

#### 4.7.2. TEAC Assay

The Trolox equivalent antioxidant capacity was evaluated according to Monforte et al. [34]. The reaction mixture consisting of 4.3 mM K_2_S_2_O_8_ and 1.7 mM ABTS solution (1:5 *v*/*v*) was incubated for 12 h in the dark at RT, diluted just before the analyses up to an absorbance of 0.7 ± 0.02 (734 nm) and used within 4 h. Fifty microliters of LE and RE (3.0–24 µg/mL and 12.5–100 µg/mL, respectively) were added to 1 mL of the reaction mixture and incubated at RT for 6 min. The absorbance was recorded at 734 nm using the same instrument and blank reported in Section 4.5.1. Trolox was used as a reference compound (0.63–5.0 µg/mL).

#### 4.7.3. FRAP Assay

Ferric reducing antioxidant power was evaluated according to Smeriglio et al. [35]. Fifty microliters of LE and RE (3.0–24 µg/mL and 12.5–100 µg/mL, respectively) were added to 1.5 mL of fresh pre-warmed (37 °C) working FRAP reagent (300 mM buffer acetate pH 3.6, 10 mM 2,4,6-Tris (2-pyridyl)-s-triazine (TPTZ)-40 mM HCl, and 20 mM FeCl_3_) and incubated for 4 min at RT in the dark. Absorbance was recorded at 593 nm using the same instrument and blank reported in Section 4.5.1. Trolox was used as a reference compound (1.25–10 µg/mL).

#### 4.7.4. ORAC Assay

Oxygen radical absorbance capacity was evaluated according to Smeriglio et al. [36]. In brief, 20 µL of LE and RE (0.25–2.0 µg/mL and 0.25–2.0 µg/mL, respectively) diluted in 75 mM phosphate buffer pH 7.4 was added to 120 µL of fresh 117 nM fluorescein and incubated 15 min at 37 °C. Sixty microliters of 40 mM AAPH radical were added to start the reaction, which was monitored every 30 s for 90 min (λ_ex_ 485; λ_em_ 520) by a fluorescence plate reader (FLUOstar Omega, BMG LABTECH, Ortenberg, Germany) against a blank superimposable to what reported in Section 4.5.1, and by using Trolox as reference compound (0.25–2.0 µg/mL).

### 4.8. Anti-Inflammatory Activity

The anti-inflammatory activity of *P. ostruthium* LE and RE was evaluated by three in vitro enzymatic and non-enzymatic assays. Absorbance was recorded by a multi-well plate reader (Multiskan GO; Thermo Scientific, Waltham, MA, USA). Results were expressed as reported in Section 4.7.

#### 4.8.1. Bovine Serum Albumin (BSA) Denaturation Assay

The ability of LE and RE to inhibit the heat-induced BSA denaturation was evaluated according to Smeriglio et al. [37]. In brief, 100 µL of 0.4 % fatty-free BSA solution and 20 µL of PBS pH 5.3 were added to a 96-well plate. Therefore, 80 µL of LE and RE (3.13–25.0 µg/mL and 12.5–100 µg/mL, respectively) were added. The absorbance was recorded at 595 nm at starting time and after incubation for 30 min at 70 °C against a blank superimposable to what was reported in Section 4.5.1. Diclofenac sodium was used as a reference compound (3.0–24 µg/mL).

#### 4.8.2. Protease Inhibition Assay

The protease inhibitory activity was evaluated according to Smeriglio et al. [38]. In brief, 200 µL of LE and RE (6.25–50.0 µg/mL and 12.5–100 µg/mL, respectively) were added to the reaction mixture consisting of 12 µL of trypsin (10 µg/mL) and 188 µL of 25 mM Tris-HCl buffer (pH 7.5). After that, 200 µL of 0.8% casein was added, the reaction mixture was incubated for 20 min at 37 °C in a water bath, then 400 µL of perchloric acid was added to stop the reaction and samples centrifuged at 3500× *g* for 10 min. The absorbance of the supernatants was recorded at 280 nm against the same blank reported in Section 4.5.1. Diclofenac sodium was used as a reference compound (2.0–16 g/mL).

#### 4.8.3. Lipoxygenase (LOX) and Cyclooxygenase (COX-2) Inhibition Assays

The effects of extracts on LOX were evaluated using the Cayman’s Lipoxygenase Inhibitor Screening Assay Kit Reagents (soybean lipoxygenase, item n. 760700, Cayman Chemical, Ann Arbor, MI, USA). The effects of extracts on COX were evaluated using the Cayman’s COX inhibitor Screening Assay (item n. 560131, Cayman Chemical) on human recombinant COX-2 (item n. 460121, Cayman Chemical). For both enzymes, the assays were conducted in 96-well plates, while protocols and calculations of percent inhibition were performed according to the manufacturer’s instructions. Nordihydroguaiaretic acid (NDGA, 100 µM) and nimesulide (3 µg/mL) were used as positive controls for LOX and COX-2 inhibition, respectively. Plates were read at 490 nm for LOX and at 412 nm for COX in a Varian Cary-50 Bio spectrophotometer (Agilent, Milan, Italy).

### 4.9. Cell Viability and Wound Healing Assays

The effects of extracts on cell viability were evaluated by the 3-(4,5-dimethylthiazol-2-yl)-2,5-diphenyltetrazolium bromide (MTT) test, while the healing activity with scratch wound assay was evaluated as previously described [39]. Both HaCat and L929 cells were grown in culture flasks at 37 °C, in 5% CO_2_, humidified atmosphere, in Dulbecco’s modified Eagle medium enriched with 10% (*v*/*v*) fetal bovine serum (EuroClone, Milan, Italy), 1% L-glutamine, and 1% antibiotic mixture (Penicillin/Streptomycin). At the reaching of confluence, the growth medium was removed, and substrate-adhering cells were washed with PBS. For HaCat cells, an additional pre-treatment with PBS-EDTA for 10 min at 37 °C was performed. Cells were then detached from the substrate by incubating for 10 min at 37 °C with trypsin (0.05%)-EDTA (0.02%) in PBS (Euroclone, Milan, Italy), and the cell suspension was collected and added to a fresh growth medium in a 1:4 ratio to block the enzymatic activity of trypsin. Aliquots of cells were stained with Turk dye and counted with a Thoma Chamber.

#### 4.9.1. Cell Viability Assay

For cell viability assays, HaCaT and L929 cells were seeded on 96-well plates (20,000 cells/well), incubated in a growth medium for 24 h, exposed to increasing concentrations of extracts for 24 or 48 h (separated tests), processed for MTT, and read at 570 nm. Absorbance data were used to obtain dose–response curves and derive IC_05_ and IC_50_ values.

#### 4.9.2. Wound Healing Test

Both HaCaT and L929 cells were seeded in 12-well plates (200,000 cells/well) and incubated in a growth medium up to when a monolayer was obtained. For each well, the cell layer was wounded twice with a sterile 200 μL pipette tip. Subsequently, cells were exposed to increasing concentrations of extracts for 24 h and then fixed with FineFIX solution and stained with TBO dye. Positive controls were obtained by incubation with 50 μg/mL allantoin (Sigma-Aldrich, 05670-25G). Wound width at time zero was recorded by fixing and staining a few wells immediately after wounding. Wounded cell layers were photographed using a Leica M205 C stereomicroscope coupled to a Leica EZ 2.1.5 camera, wound widths were measured by the ImageJ software (https://imagej.nih.gov/ij/, accessed on 28 June 2022), and wound closure was expressed as a percentage of control closure [40].

## 5. Conclusions

Our study provides new information on the phytochemical composition and medicinal properties of *P. ostruthium*, an alpine species mainly known since ancient times for the traditional use of its rhizome. Considering that in some areas of the Alps, the leaf is also used to treat skin problems, we investigated and compared the hydroalcoholic extracts of the two plant parts. The LE proved to be the most promising extract due to its antioxidant and anti-inflammatory properties. Both extracts showed low cytotoxicity on human keratinocytes and on murine fibroblasts and significant wound healing activity, leading to the validation of the traditional uses of this species for the treatment of skin diseases and inflammatory conditions. In addition, the low presence of furanocoumarins in the LE makes it an ideal candidate to be included in dermatological and cosmeceutical products.

## Figures and Tables

**Figure 1 molecules-27-04271-f001:**
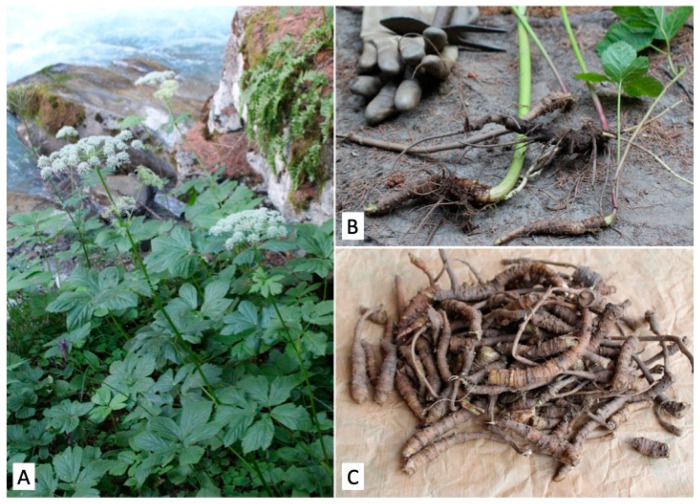
(**A**) Plants of *P. ostruthium* in their natural environment. (**B**) Freshly sampled rhizomes. (**C**) Clean rhizomes used for micromorphological analysis and extract preparation.

**Figure 2 molecules-27-04271-f002:**
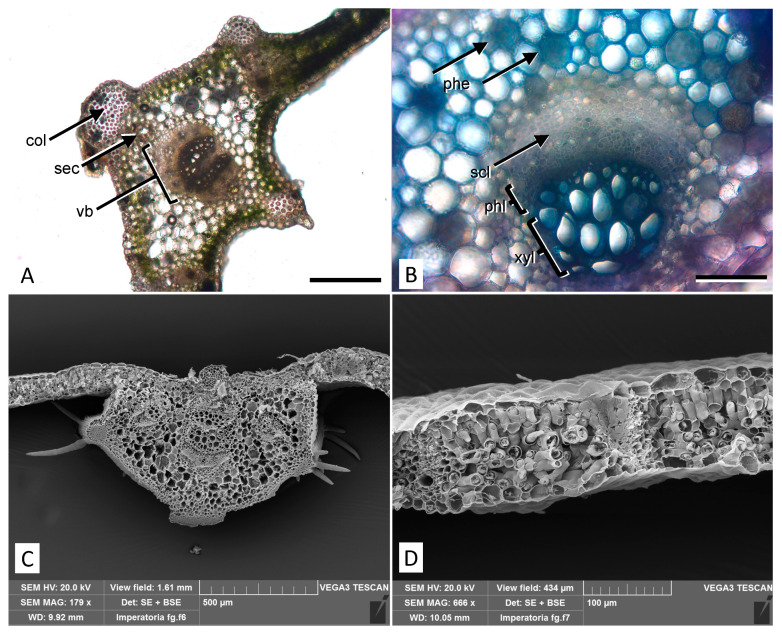
Micromorphological features of the *P. ostruthium* leaf. (**A**) Light microscopy view of a cross section at the midvein level with collenchyma protrusions. col: collenchyma; sec: secretory channel; vb: vascular bundle. Bar 500 µm. (**B**) Detail of a vascular bundle after staining with TBO: around the bundle, several parenchymatous cells rich in polyphenols appear blue/green (phe). scl: sclerenchyma; phl: phloem; xyl: xylem. Bar 200 µm. (**C**) SEM micrograph of a cross section at the midvein level showing typical trichomes located on the collenchyma protrusions. (**D**) SEM view of a leaf cross section showing the mesophyll structure.

**Figure 3 molecules-27-04271-f003:**
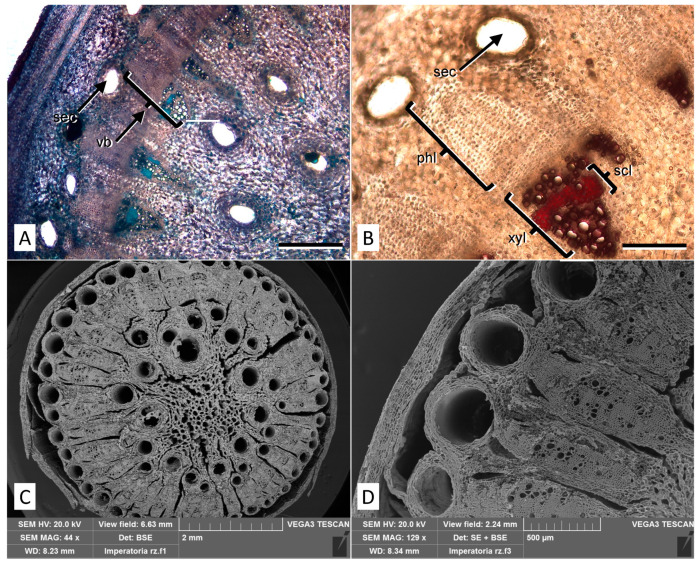
Micromorphological features of the *P. ostruthium* rhizome. (**A**) Light microscopy view of a cross section stained with TBO: different vascular bundles (vb) and secretory channels (sec) are visible. Bar 500 µm. (**B**) Detail of a cross-section after phloroglucinol-HCl staining, showing xylem vessels (xyl) interspersed with sclerenchyma fibers stained in purple/red (scl). sec: secretory channel; phl: stratified phloem. Bar 200 µm. (**C**) SEM micrograph of a cross section of an old rhizome, where the high number of vascular bundles and secretory channels are highlighted. (**D**) Detail of (**C**) at higher magnification.

**Figure 4 molecules-27-04271-f004:**
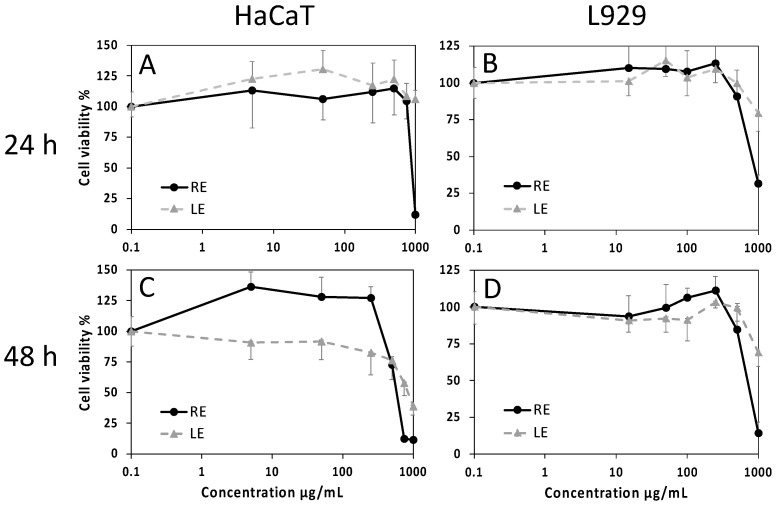
Cell viability evaluated by the MTT assay on HaCaT human keratinocytes (**A**,**C**) and L929 mouse fibroblasts (**B**,**D**) exposed for 24 h (**A**,**B**) or 48 h (**C**,**D**) to increasing concentrations of LE or RE. Data are mean absorbances ± S.D. of 570 nm readings, obtained from 6 replicate wells for each condition in two independent experiments. Values of IC_50_ and IC_05_ at 48 h are reported in Table 5.

**Figure 5 molecules-27-04271-f005:**
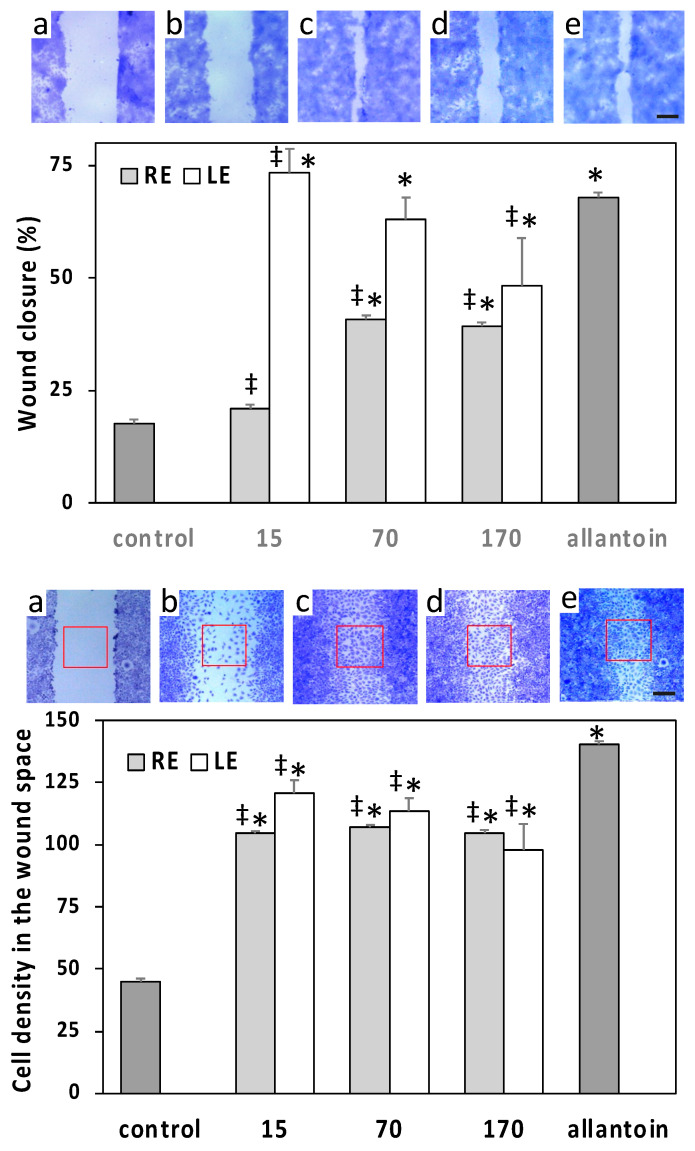
Scratch wound assays conducted on HaCaT human keratinocytes (top panel) and L929 mouse fibroblasts (bottom panel) exposed to different concentrations of LE or RE (µg/mL). Micrographs show representative wounded cell monolayers at different times after wounding and under different conditions (bar = 250 µm). Data of HaCaT are expressed as means ± S.D. (*n* = 49–67) of percent wound closure at 24 h. Data of L929 are means ± S.D. (*n* = 8) of the normalized cell densities within the wound space (number of cells per unit area; red squares in micrographs) at 24 h post wounding. a = wound space just after scratch wounding; b = control at 24 h; c = 15 µg/mL LE at 24 h; d = 70 µg/mL RE at 24 h; e = positive control allantoin at 24 h; * = significantly different from control; ‡ = significantly different from allantoin (*p* < 0.01).

**Table 1 molecules-27-04271-t001:** Phytochemical screening of *P. ostruthium* leaf and rhizome hydroalcoholic extracts (LE and RE, respectively). Results are the mean ± standard deviation (S.D.) of three independent experiments in triplicate (*n* = 3).

Assay	LE	RE
Total phenols (mg GAE ^a^/100 g DE ^b^)	10,668.30 ± 581.55	9538.00 ± 622.24
Flavonoids (mg RE ^c^/100 g DE)	52,914.94 ± 384.84 *	13,694.83 ± 561.33
Flavan-3-ols (mg CE ^d^/100 g DE)	200.19 ± 1.58 *	334.89 ± 12.66
Proanthocyanidins (mg CyE ^e^/100 g DE)	0.078 ± 0.00 *	0.003 ± 0.00
Polimerization index ^f^	2575.46 *	111.630

^a^ GAE, Gallic acid equivalents; ^b^ DE, Dry extract; ^c^ RE, Rutin equivalents; ^d^ CE, Catechin equivalents, ^e^ CyE, Cyanidin equivalents; ^f^ Polymerization index = Flavonols/Proanthocyanidins. * *p* < 0.005 vs. RE.

**Table 2 molecules-27-04271-t002:** Major components identified and quantified in *P. ostruthium* LE and RE by LC-DAD-ESI-MS analysis.

	Compound	[M-H]^−^	[M-H]^+^	λ_max_	LE	RE
		(*m*/*z*)	(*m*/*z*)	(nm)	mg/100 g DE	
1	3-*O*-Caffeoylquinic acid	353	355	296, 326	4.05 ± 0.14 *	195.55 ± 1.67
2	5-*O*-Caffeoylquinic acid	353	353	296, 326	5.14 ± 0.05 ^a,^*	114.36 ^a^ ± 1.44
3	4-*O*-Caffeoylquinic acid	353	355	296, 326	135.0 ± 0.57 ^a,^*	16.28 ^a^ ± 0.08
4	5-*O*-p-Coumaroylquinic acid	337	339	296, 324	3.70 ± 0.02 ^b^	-
5	5-*O*-Feruloylquinic acid	367	369	296, 324	5.05 ± 0.08 ^c,^*	59.80 ^c^ ± 0.05
6	4-*O*-Feruloylquinic acid	367	369	296, 324	-	1.93 ^c^ ± 0.01
7	p-Coumaroyl glucose	325	327	226, 315	0.13 ± 0.00 ^b^	-
8	Quercetin-3-*O*-rutinoside	609	611	257, 354	95.28 ± 0.84	-
9	3,4-di-*O*-Caffeoylquinic acid	515	517	296, 324	11.08 ± 0.05 ^a,^*	2.69 ^a^ ± 0.02
10	Hesperidin	609	611	284, 332	-	9.57 ± 0.06
11	Quercetin-3-*O*-(6″acetyl-glucoside)	505	507	256, 356	138.52 ± 1.88 ^d^	-
12	3,7-Dimethylquercetin	329	331	257, 358	2.28 ± 0.03 ^e^	-
13	Oxypeucedanin-hexoside	465	467	313	46.98 ± 0.42 ^f,^*	0.43 ^f^ ± 0.01
14	Kaempferol 3-*O*-acetyl-glucoside	489	491	265, 328	501.24 ± 0.66 ^g^	-
15	Osthenol-7-*O*-glucoside	-	393	270, 320	-	0.73 ^h^ ± 0.01
16	Oxypeucedanin-malonyl-hexoside	-	553	270, 315	-	0.14 ^f^ ± 0.00
17	Oxypeucedanin hydrate	-	305	311	1.14 ± 0.01 ^f,^*	5.67 ^f^ ± 0.04
18	Oxypeaucedanin 2′-acetate-3′glucoside	-	509	311	3.25 ± 0.02 ^f,^*	2.13 ^f^ ± 0.01
19	Oxypeucedanin	-	287	309	5.81 ± 0.02 ^f,^*	3.05 ^f^ ± 0.03
20	Oxypeucedanin ethanolate	-	333	311	8.62 ± 0.04 ^f^	-
21	Ostruthol	-	387	309	-	1.45 ± 0.02
22	Isoimperatorin	-	271	300	-	29.55 ± 0.08
23	Imperatorin	-	271	310	-	7.31 ± 0.05
24	Ostruthin	-	299	330	-	281.88 ± 2.24

Percentage distribution (%) of phytochemical classes						
Phenolic acids					16.97	53.32
Flavonoids					76.23	1.31
Coumarins					6.80	45.37

Data are the mean ± standard deviation (S.D.) of three independent experiments in triplicate (*n* = 3), expressed as mg/100 g dry extract (DE). Quantification was carried out by building external calibration curves of reference standards, whereas the superscript letters (a–h) indicate that the quantification was carried out based on the calibration curves of the following structural analogues: 3-*O*-Caffeoylquinic acid, Coumaric acid, Ferulic acid, isoquercetin, quercetin, oxypeucedanin, Kaempferol 3-*O*-glucoside, and Osthenol, respectively; * *p* < 0.005 vs. RE.

**Table 3 molecules-27-04271-t003:** Determination of antioxidant and anti-inflammatory activities of LE and RE by several in vitro colorimetric assays based on different environment and reaction mechanisms.

Assay	LE	RE	Reference Standard ^b^
Antioxidant activities			
2,2-Diphenyl-1-picrylhydrazyl (DPPH)	24.11 (20.14–28.87) *	152.73 (61.54–379.06)	8.57 (4.88–10.22) §
Trolox equivalent antioxidant capacity (TEAC)	12.14 (10.34–14.24) *	51.07 (35.56–73.35)	4.89 (2.24–6.95) §
Ferric reducing antioxidant power (FRAP)	19.37 (15.59–24.07) *	47.18 (39.75–56.01)	5.38 (3.86–8.01) §
Oxygen radical absorbance capacity (ORAC)	1.03 (0.76–1.40)	1.35 (1.09–1.69)	0.72 (0.38–0.92) §
Anti-inflammatory activities			
BSA ^a^ denaturation assay	15.16 (12.97–17.72) *	57.06 (47.72–69.70)	17.58 (15.05–19.68) °
Protease inhibitory activity	24.78 (19.75–31.09)	30.04 (23.42–38.51)	6.88 (3.26–9.44) §

Data are half-maximal inhibitory concentrations (IC_50_, µg/mL) with confidence limits (C.L.) derived from three independent experiments in triplicate. ^a^ BSA, Bovine serum albumin; ^b^ Reference standards: Trolox for DPPH, TEAC, FRAP and ORAC assays; Diclofenac sodium for BSA and protease inhibitory activity assay. * *p* < 0.005 vs. RE; § *p* < 0.005 vs. LE and RE; ° *p* < 0.005 vs. RE.

**Table 4 molecules-27-04271-t004:** In vitro inhibition of LE and RE on cyclooxygenase (COX-2) and lipoxygenase (LOX) enzymatic activities.

Enzyme	LE		RE		Standard
	150 µg/mL	300 µg/mL	150 µg/mL	300 µg/mL	
COX-2	67.3 ± 11.5	43.8 ± 4.4	n.d.	n.d.	87.9 ± 0.1 *
LOX	52.0 ± 27.3 ‡	78.7 ± 8.8 #	11.3 ± 11.3	65.4 ± 13.5 #	96 ± 3.5 *

Data are mean percent inhibition ± S.D. (*n* = 3). Reference standard: 3 µg/mL nimesulide for COX-2, and 100 µM nordihydroguaiaretic acid for LOX. ‡ *p* < 0.05 in a *t*-test comparison between different concentrations of the same extract; # *p* < 0.05 in a *t*-test comparison between different extracts at the same dose; * *p* < 0.05 with respect to all other groups, n.d. = not detectable.

**Table 5 molecules-27-04271-t005:** Median (IC_50_) and threshold (IC_05_) concentrations of LE and RE on HaCaT human keratinocytes and L929 mouse fibroblasts evaluated by the MTT assay at 48 h endpoints.

	Leaf Extract		Rhizome Extract	
	IC_50_	IC_05_	IC_50_	IC_05_
HaCaT	897 (760–1059)	252 (113–559)	439 (416–463)	364 (293–451)
L929	1094 (607–1970)	801 (189–3394)	681 (603–768)	385 (320–462)

Data are expressed as extract concentrations in µg/mL. Values of 95% confidence intervals are shown in parentheses.

## Data Availability

Not applicable.

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
