# Peer review of "Anti-Inflammatory and Wound Healing Properties of Leaf and Rhizome Extracts from the Medicinal Plant Peucedanum ostruthium (L.) W. D. J. Koch"

_molecules, 2022, doi:10.3390/molecules27134271_

Round 1

Reviewer 1 Report

Despite the traditional medicinal uses of the rhizome, the results showed that the leaves also possess an equal or greater ability to heal superficial wounds. Authors demonstrate that the leaves can be used for a variety of other purposes as well. I think the data are not sound and this manuscript cannot be published in "Molecules " in this version. My comments are described followings:

Comments:

1. Authors suggested that LE or RE has an antioxidant and anti-inflammatory activities. However, the key mechanism or markers must be validated by western blot in the revised version.

2. Please describe how to culture the HaCaT keratinocytes and L929 fibroblasts. Additionally, please describe how many cell-culture passages were studied. If this is the case, how many replicates were performed in each passage? Please include this information in the manuscript, either in Methods or Legend to Figures.

3. Please study the cytotoxic effect of LE and RE on HaCaT human 274 keratinocytes and L929 mouse fibroblasts in a time dependent manner.

4. The action mechanism in this study must be explain.

5. For scratch wound assays, authors must add the cytosine β - D -arabino furanoside to block the cell proliferation for 4 h firstly.

6. The authors should consult an English language editor and write in an "academic English" as well as "must" provide proof of a certificate of editing before publishing.

7. I encourage authors to provide the graphic summary in the revised version.

Author Response

Reviewer in Roman, Authors in Italics

1.Authors suggested that LE or RE has an antioxidant and anti-inflammatory activities. However, the key mechanism or markers must be validated by western blot in the revised version.

The aim of this study is to provide a confirmation to popular uses that suggests therapeutic properties of the plant for skin problems. In addition, the properties of the leaf extract have been studied since the aerial portions of the plant have not been considered to date as an herbal remedy. Results are based on the cumulative effects of the phytocomplex, while for a mechanistic approach the use of isolated compounds would be more suitable, and moreover, the investigation of possible targets by e.g. WB would require preliminary high throughput analyses like proteomics or NA microarrays. As above stated, these investigations are beyond the aims of the present study and will be the argument of further research.

2.Please describe how to culture the HaCaT keratinocytes and L929 fibroblasts. Additionally, please describe how many cell-culture passages were studied. If this is the case, how many replicates were performed in each passage? Please include this information in the manuscript, either in Methods or Legend to Figures.

More detailed cell culture protocols have been reported in the Methods, including the number of passages of cell batches used in experiments. The number of replicates for each experiments are reported in the text, figure captions, and tables.

3.Please study the cytotoxic effect of LE and RE on HaCaT human 274 keratinocytes and L929 mouse fibroblasts in a time dependent manner.

The results of cytotoxicity assays with two endpoints, 24 and 48 h, have now been reported.

4.The action mechanism in this study must be explain.

Please see the reply to point n. 1.

5.For scratch wound assays, authors must add the cytosine β - D -arabino furanoside to block the cell proliferation for 4 h firstly.

Wound healing depends essentially on two processes, cell migration and proliferation. Our study was aimed at measuring wound healing, not simply cell migration, and therefore, blocking cell proliferation would have conducted to biased results.

6.The authors should consult an English language editor and write in an "academic English" as well as "must" provide proof of a certificate of editing before publishing.

The manuscript has been revised by a qualified language editor.

7.I encourage authors to provide the graphic summary in the revised version.

A graphical abstract has been included in the re-submission.

Reviewer 2 Report

The main target of the manuscript was to provide experimental confirmations of the therapeutic properties 
of P. ostruthium. However, there is a lack of information about the extraction of bioactive compounds from leaves and rhizomes and how they were optimized. Indicates one only the reference bibliographic is not enough. The scalability of the process and crop domestication of the plant aspects should be included in the conclusion.

Author Response

Reviewer in Roman, Authors in Italics

The main target of the manuscript was to provide experimental confirmations of the therapeutic properties of P. ostruthium. However, there is a lack of information about the extraction of bioactive compounds from leaves and rhizomes and how they were optimized. Indicates one only the reference bibliographic is not enough.

We thank the Reviewer for this comment. The manuscript has been revised accordingly and specifications regarding the fine-tuning and optimization of the extraction procedure adopted in this study have been added to the Materials and Methods in section 4.4.

The scalability of the process and crop domestication of the plant aspects should be included in the conclusion.

The relevance of our data for the scalability of plant extract production has been reported at the end of the Discussion.

Reviewer 3 Report

The comments are attached

Author Response

Reviewer in Roman, Authors in Italics

The manuscript is interesting however some problems are detected in the methodology and results. Also, the manuscript is not so novelty, since there are some similar studies:-Nani M, Leone A, Bom VP, et al. Evaluation and Comparison of Wound Healing. Properties of an Ointment (AlpaWash) Containing Brazilian Micronized Propolis and Peucedanum ostruthium Leaf Extract in Skin Ulcer in Rats. International Journal of Pharmaceutical Compounding. 2018 Mar-Apr;22(2):154-163. PMID: 29877862. -Identification and Quantification of Coumarins in Peucedanum ostruthium (L.) Koch by HPLC-DAD and HPLC-DAD-MS. Sylvia Vogl, Martin Zehl, Paolo Picker, Ernst Urban, Christoph Wawrosch, Gottfried Reznicek, Johannes Saukel, and Brigitte Kopp. Journal of Agricultural and Food Chemistry 2011 59 (9), 4371-4377. DOI: 10.1021/jf104772x -Lammel, C.; Zwirchmayr, J.; Seigner, J.; Rollinger, J.M.; de Martin, R. Peucedanum ostruthium Inhibits E-Selectin and VCAM-1 Expression in Endothelial Cells through Interference with NF-κB Signaling. Biomolecules 2020, 10, 1215. 

The aim of this study is to provide a confirmation to popular uses that suggest therapeutic properties of the plant for skin problems. In addition, in this study the properties of the leaf extract have been studied since the aerial portions of the plant have not been considered to date as an herbal remedy. The articles indicated by the reviewers have different topics and therefore do not affect the novelty of our findings. The first two articles were already quoted in the text, while the third one has been included in the revised version.

Other aspects to take into account would be related to the replications. Did authors consider enough 3 replications ?

A total of three replicates is generally considered suitable in biochemical assays at the CV levels shown by our data. In cell-based experiments the number of replicates is always higher than 3. This was not clearly expressed for cytotoxicity data and this point has been now amended in the caption to Fig. 4.

Round 2

Reviewer 1 Report

Many thanks for authors' rebuttal. I have no more questions.